

# Integration of temporal & spatial properties of dynamic functional connectivity based on two-directional two-dimensional principal component analysis for disease analysis

Feng Zhao[1], Ke Lv[1], Shixin Ye[1], Xiaobo Chen[1], Hongyu Chen[2], Sizhe Fan[3], Ning Mao[4] and Yande Ren[5]

[1] School of Computer Science and Technology, Shandong Technology and Business University, Yantai, China
[2] School Hospital, Shandong Technology and Business University, Yantai, China
[3] Canada Qingdao Secondary School (CQSS), Qingdao, China
[4] Department of Radiology, Yantai Yuhuangding Hospital, Yantai, China
[5] Department of Radiology, The Affiliated Hospital of Qingdao University, Qingdao, China

Corresponding author
Hongyu Chen,
chenhongyu1016@163.com

## ABSTRACT

Dynamic functional connectivity, derived from resting-state functional magnetic resonance imaging (rs-fMRI), has emerged as a crucial instrument for investigating and supporting the diagnosis of neurological disorders. However, prevalent features of dynamic functional connectivity predominantly capture either temporal or spatial properties, such as mean and global efficiency, neglecting the significant information embedded in the fusion of spatial and temporal attributes. In addition, dynamic functional connectivity suffers from the problem of temporal mismatch, *i.e.*, the functional connectivity of different subjects at the same time point cannot be matched. To address these problems, this article introduces a novel feature extraction framework grounded in two-directional two-dimensional principal component analysis. This framework is designed to extract features that integrate both spatial and temporal properties of dynamic functional connectivity. Additionally, we propose to use Fourier transform to extract temporal-invariance properties contained in dynamic functional connectivity. Experimental findings underscore the superior performance of features extracted by this framework in classification experiments compared to features capturing individual properties.

## INTRODUCTION

The human brain is a complex functional system that is artificially divided into different brain regions. Each brain region has a different primary function and they work together to accomplish complex tasks. An in-depth study of the interactions between brain regions can reveal the operating mechanisms of the brain and lay the foundation for the discovery of the pathogenesis of brain diseases (*Angermann et al., 2022*; *Bai et al., 2009*; *Chopade*

*et al., 2023*). In order to study the correlations between different brain regions of the brain as they work, a large number of studies have used resting-state functional magnetic resonance (rs-fMRI) to construct functional connectivity (FN) between different brain regions, which characterizes the interactions between different brain regions (*Damaraju et al., 2014*; *Dawson, Rieder & Johnson, 2023*; *Farah et al., 2020*; *Gao et al., 2022*; *Grandjean et al., 2023*). Currently, functional connectivity is now a key tool to help further our understanding and aid in the diagnosis of neurological disorders such as Parkinson's disease and autism (*Holz et al., 2023*; *Huber et al., 2021*; *Hutchison et al., 2013*; *Kazeminejad & Sotero, 2019*).

Functional connectivity is generally categorized into two main groups: (1) static functional connectivity, which assumes that rs-fMRI is constant throughout the recording period and does not change over time. Higher-order brain networks based on static functional connectivity achieve good results in assisting the diagnosis of neurological diseases (*Kumar & Aravind, 2010*; *Ladwig et al., 2022*; *Lawrence et al., 2020*; *Li et al., 2020*); (2) dynamic functional connectivity (DFC), which characterizes the dynamics of functional connectivity over time, and is generally derived from functional connectivity time series by the "sliding window" method. Dynamic functional connectivity has better performance on classification tasks because it contains time-varying information (*Liu et al., 2020*; *Liu et al., 2019*; *Long et al., 2020*; *Lu et al., 2019*; *Lurie et al., 2020*; *Matson & Nebel-Schwalm, 2007*). On the one hand the dynamics of functional connectivity over time can be referred to as the temporal properties of dynamic functional connectivity. On the other hand, the brain as a complex functional system has not only interactions between regions of interest (ROIs), but also complex spatial relationships between different functional connections, which reflect the spatial properties of functional connections.

Although dynamic functional connectivity includes both temporal and spatial properties, researchers have mostly analyzed one aspect of a single property. For example, for temporal properties, most studies use statistical methods to count the minimum, maximum, mean, standard deviation, root mean square, *etc.*, of the time series of dynamic functional connectivity (*Ni et al., 2014*; *Olczyk et al., 2022*; *Pang et al., 2022*). For spatial properties, most researchers have represented different functional brain regions and their connections as a brain network, and then statistically analyzed the characteristic path lengths, global efficiencies, and clustering coefficients of this network (*Park et al., 2020*; *Qiu et al., 2014*; *Razzak et al., 2020*). Few studies have been conducted to extract and analyze features that integrate both temporal and spatial properties of dynamic functional connectivity (*Ricaud et al., 2019*). It can be reasonably assumed that features combining complementary temporal and spatial properties can provide more critical information for brain disease research and more differential information for classifiers, which in turn improves the correctness of the diagnosis of neurological disorders and the diagnostic efficiency of doctors. Therefore, a framework is needed to extract features that simultaneously contain the temporal and spatial properties of dynamic functional connectivity.

Based on the above analysis, we propose a new feature extraction framework based on two-directional two-dimensional principal component analysis ((2D)$^2$PCA) (*Royer et al., 2022*; *Sadat & Joye, 2020*; *Shen et al., 2023*), which is capable of extracting features that

integrate both temporal and spatial characteristics of dynamic functional connectivity. In this framework, we utilize the $(2D)^2PCA$ function that can fully consider the properties of the matrix in both directions while eliminating noise, $(2D)^2PCA$ is used to simultaneously consider the relationship between the temporal and spatial orientations of the dynamic functional connectivity, and thus integrates the properties in both directions. To the best of our knowledge, there is no $(2D)^2PCA$ based feature extraction method for dynamic functional connectivity to extract features. In contrast to previous feature extraction methods that only capture structural information in a single direction, the proposed $(2D)^2PCA$ based feature extraction method can simultaneously integrate features in both directions.

In addition, dynamic functional connectivity suffers from the problem of sensitive to the chronological order of its subnetworks, which limits its use in comparative studies (*Shi et al., 2019*). The Fourier transform has been widely used in the field of communications, acoustics and optics, where it converts signals that are difficult to analyze in the time domain into a frequency domain representation and then goes on to analyze them (*Starck et al., 2013*; *Sun et al., 2022*; *Sun et al., 2019*; *Valentine, Al-Mualem & Baiz, 2021*; *Valsasina et al., 2019*). Inspired by the Fourier transform function, we propose to use the Fourier transform to solve the problem. In the feature extraction framework, the time series of different functional connections are converted from time domain to frequency domain representation by Fourier transform. By comparing the information on the same frequency instead of comparing the information on the same time point, the impact of temporal mismatch on to the dynamic functional connectivity comparison study is mitigated.

In order to verify that the features extracted by the framework have better separability, we trained classifiers using the extracted features for subsequent adjunctive diagnosis of neurological disorders and conducted experiments in two brain disease classification tasks. The experimental results show that the features extracted by the framework perform better in disease-assisted diagnosis than those extracted by feature extraction methods that only include temporal or spatial unidirectional features, and the Fourier transform mitigates the problem of dynamic functional connectivity being sensitive to the temporal order of its subnetworks. In addition, the extracted features show a performance improvement over the comparison methods on both classification tasks, and it can be found that our proposed framework is aiding in the diagnosis of neurological disorders, and not only for a particular disease.

In summary, there are two parts of contribution in this article: (1) proposing a new feature extraction framework for extracting features that integrate temporal and spatial properties of dynamic functional connectivity; (2) utilizing the Fourier transform method to capture dynamic functional connectivity properties without performing chronological time matching. The framework has promising applications in the detection, prevention and prognostic assessment of brain diseases, and the abnormal functional connectivity and brain regions of brain disease patients identified in the article can provide new perspectives for subsequent research.
## MATERIALS AND DATA PREPROCESSING

### Subjects

The rs-fMRI data of 45 autism spectrum disorders (ASD) patients and 47 healthy controls (HC) groups were scanned using a 3-T Siemens Allegra scanner at the NYU Langone Medical Center. Due to the difference in medical device, collection protocol, *etc* to mitigate data heterogeneity, scans were exclusively considered from 45 individuals diagnosed with ASD and 47 individuals within the age range of 7 to 15 years. The imaging was conducted at NYU Langone Medical Center. All subjects under consideration exhibited minimal head movement, with displacement <1.5 mm or angular rotation <1.5° in any of the three directions. The ASD subjects were diagnosed according to the autism criteria in the Diagnostic and Statistical Manual of Mental Disorders (4th Edition, Text Revised) (DSM-IV-TR) (*American Psychiatric Association, 2000*). For more details of the data collection, exclusion criteria, and scan parameters, please visit the ABIDE website. The main scanning parameters used in this dataset include the flip angle = 90, 33 slices, TR/TE = 2,000/15 ms, 180 volumes, and voxel thickness = 4 mm.

The rs-fMRI data of 52 end-stage renal disease (ESRD) patients and 49 HC groups were scanned using GE Signa HDX 3.0T MRI equipment at hospital of Qingdao University. Inclusion criteria for the ESRD group: (1) Diagnosed by the Affiliated Hospital of Qingdao University as Chronic Kidney Disease Stage 5, glomerular filtration rate less than 15 ml $\cdot$min$^{-1}\cdot\left(1.73\ m^2\right)^{-1}$. (2) Age 18–70 years old; (3) No contraindications to magnetic resonance examination and claustrophobia. Exclusion criteria: (1) history of severe traumatic brain injury; (2) intracranial organic pathology, intracranial organic disease, such as tumour, infarction, haemorrhage, *etc.*; (3) cerebrovascular disease, such as cerebral arteriovenous malformation, smoky disease, *etc.* and smog disease; (4) history of mental illness; and (5) history of substance abuse (drug ethanol or cigarettes); (6) hypertension, diabetes mellitus, coronary artery disease, heart failure, liver and kidney failure, *etc.* failure, liver or kidney failure, and other serious systemic diseases; (7) incompatibility with the test subjects who are incompatible with the test or unable to effectively complete the MRI scan. The scan parameters are TR = 3,000 ms, TE = 40 ms, FA = 90°, 25 slices, thickness = 5 mm, gap = 0 mm, matrix size = 96×96 and FOV = 24 cm × 24 cm.

### Data preprocessing

Data preprocessing is performed for all rs-fMRI data by using a standard pipeline, including time correction, head motion correction, spatial normalization and smoothing using MATLAB's DPABI toolbox. Specifically, (1) conversion of data format; (2) removal of data from the first 10 sampling points for each subject in order to exclude the effects of initial magnetic field instability and the subject's initial emotional instability; (3) correction of time layer, head motion. Subjects with excessive head movements are excluded; (4) spatial normalization in terms of MNI standard space with the resolution of $3 \times 3 \times 3$ mm$^2$; (5) ventricle signals, cerebrospinal fluid signal and white matter signal were regressed out as nuisance signals; (6) to filter out various physiological noises to reduce the effects of low-frequency linear drift and physiological noise, the data were processed to remove linear trends and band-pass filtered (0.01−0.08 Hz).

In this experiment, after data pre-processing, the brain was divided into 116 brain regions for each subject according to the AAL template, and the average of the BOLD signal of all voxels in each brain region was calculated to obtain the average time series of the 116 brain regions.

## METHODS

In this section, we first introduce DFC construction, the Fourier transform and $(2D)^2$PCA technology, and then present the details of the construction of the framework proposed in this article.

### Construction of DFC

For each subject, let $z_i = (z_{i1}, z_{i2}, \ldots, z_{iN})$ $(i = 1, 2, \ldots, 116)$ denotes the average rs-fMRI time series of the ith brain region, where N denotes the number of image time points. Dynamic FC is implemented using a sliding window approach, assuming that the length of the sliding window is T, and the step length between every two adjacent windows is S. Then the time series of length N is divided into K partially overlapping subseries with each other, where $K = \frac{[(N-T)]}{S} + 1$.

Letting $z_i(k) = [z_{i1}(k), z_{i2}(k), \ldots, z_{iT}(k)]$ $(k = 1, 2, \ldots, K)$ denotes the kth time subseries of $z_i$, then just need to list all sliding windows of FC. FC is often modeled as a FC network (FCN), with each brain ROI as a node in the network, and the strength of FC between a pair of brain ROIs as an edge. The FCN of the kth time subseries can be generated by a symmetric submatrix $D(k)$, defined as:

$$D(k) = [\rho_{ij}(k)]_{1 \le i,j \le K} \tag{1}$$

where $\rho_{ij}(k)$ denotes the Pearson's correlation between the average time subseries $z_i(k)$ and $z_j(k)$, defined as:

$$\rho_{ij}(k) = \frac{\sum_{t=1}^{T} \left(z_{it}(k) - \overline{z_i(k)}\right)\left(z_{jt}(k) - \overline{z_j(k)}\right)}{\sqrt{\sum_{t=1}^{T} \left(z_{it}(k) - \overline{z_i(k)}\right)^2} \sqrt{\sum_{t=1}^{T} \left(z_{jt}(k) - \overline{z_j(k)}\right)^2}} \tag{2}$$

where $\overline{z_i(k)}$ and $\overline{z_j(k)}$ denote average value of the average time subseries $z_i(k)$ and $z_j(k)$. The submatrix series $\{D(k)\}_{k=1}^{K}$ can describe the temporal change of the connectivity strength for all ROI pairs.

### The Fourier transform

Discrete Fourier Transform, as a mathematical tools, is commonly used in signal processing. The Discrete Fourier Transform can convert a signal into a linear combination of a set of sinusoidal functions of different frequencies, which have the property of being easy to implement and observe.

The Discrete Fourier Transform of vector $x(k), k = \{1, 2, \ldots, K\}$ is defined as:

$$f(u) = \sum_{k=1}^{K} x(k) W_K^{(k-1)(u-1)} \tag{3}$$

where $u$ denote the frequency-domain pixel coordinate of $C_r$, $u = \{1, 2, \ldots, K\}$, $W_K$ is defined as:

$$W_K = e^{\frac{-2\pi i}{K}}. \tag{4}$$

## $(2D)^2$ PCA

$(2D)^2$PCA is a matrix feature extraction technique based on two-dimensional principal component analysis (2DPCA). 2DPCA can extract whole-matrix features by finding optimal projection directions from the covariance matrix. First introduce how 2DPCA learns an optimal projection matrix reflecting the information between rows of matrixes. Letting $A \in R^{m \times n}$ be a matrix, $X \in R^{n \times d}$ be a projection matrix with n projection directions. Projecting A onto X to obtain the matrix $Y = AX$. Suppose that there are M training matrixes, denoted by $A_k (k = 1, 2, \ldots, M)$, and denote the covariance matrix as:

$$X = \frac{1}{M} \sum_{k}^{M} (A_k - \overline{A})^T (A_k - \overline{A}) \tag{5}$$

where $\overline{A}$ be denoted as:

$$\overline{A} = \frac{1}{M} \sum_{k}^{M} A_k \tag{6}$$

The optimal value for the projection matrix $X$ is composed by the vectors $X_1, X_2, \ldots, X_d$ of G corresponding to the $d$ largest eigenvalues. The value of $d$ can be controlled as follows:

$$\frac{\sum_{i=1}^{d} \lambda_i}{\sum_{i=1}^{n} \lambda_i} \geq \theta \tag{7}$$

where $\lambda = \{\lambda_1, \lambda_2, \ldots, \lambda_n\}$ is a sequence of eigenvalues in ascending order for G and $\theta$ is a pre-set threshold.

2DPCA also can learn an optimal matrix Z reflecting the information between columns, and then projects A onto Z to obtain the matrix $B = Z^T A$, and denote the covariance matrix as:

$$Z = \frac{1}{M} \sum_{k}^{M} (A_k - \overline{A})(A_k - \overline{A})^T \tag{8}$$

$(2D)^2 PCA$ is a way to simultaneously use the projection matrices X and Z. Projecting A onto X and Z, yielding a matrix C:

$$C = Z^T AX \tag{9}$$

The matrix C in Eq. (9) is the matrix after projection.

### Comparison methods

In this section we describe two comparison methods used in subsequent experiments: the central moment method and the topological index method.

### The central moment method

The feature extraction method based on the central moment method is a feature extraction method designed for retaining the temporal information of dynamic functional connections. As follows, $\rho_{ij}(k)$ is calculated according to Eq. (2), and then $d$th order central-moment $m_{ij}(d)$ of $\rho_{ij}(k)$ is calculated:

$$m_{ij}(d) = \sqrt[d]{\frac{\sum_{k=1}^{K}\left[\rho_{ij}(k) - \overline{\rho}_{ij}\right]^{d}}{K}} \quad (d = 1, 2, \ldots, D) \tag{10}$$

where D denotes the highest order, $\overline{\rho}_{ij}$ is the mean of $\rho_{ij}(k)\,(k = 1, 2, \ldots, K)$.

Different central moment features represent different information about the dynamic functional connectivity in the time direction, and it represents different discriminatory power, so the order of the central moment should be carefully chosen. $d$ is a parameter in the central moment feature framework that needs to be adjusted in subsequent training. Finally, the selected features are fed into the SVM classifier.

### The topological index method

Converting dynamic functional connectivity to topology. Specifically, first define the threshold value $T$ and ignore this functional connection by changing the value $\rho_{ij}(k) < T$ to 0. Then the topology is constructed by retaining the weights of the edges of $\rho_{ij}(k) \geq T$ or changing them to 1. We have empirically changed the top 20 per cent of weights to 1 and the rest to 0 (*Van Den Heuvel & Pol, 2010*). The calculated topological index consisted of measures of segregation (Clustering Coefficient, Transitivity), integration (Characteristic Path Length, Efficiency), and centrality (Betweenness centrality, within module degree $Z$-score, Participation coefficient) of the brain network. Formulas for each metric are presented in the article (*Wang, Zhang & Qiao, 2023*). The selected features are fed into the SVM classifier.

## Construction of the proposed framework

In this subsection, we describe the proposed framework building process in detail. Figure 1 depicts the main steps of the feature extraction framework, and it can be seen that there are four main steps: (1) Construction of the dynamic network. (2) Construction of the new network. (3) $(2D)^2 PCA$ learns the best projection matrix. (4) Projecting the new network and selecting features.

Specifically, we first construct the DFC $\{D(k)\}_{k=1}^{K}$ on the continuous rs-fMRI time series by the sliding-window strategy. Then in order to retain both temporal and spatial dynamic information, we transform the 3D DFC into a 2D new network, so that the spatial and temporal information to be stored in the row and column structure of the new network.

Figure 2 illustrates the process of new network construction. For each subject, we transform the submatrix series $\{D(k)\}_{k=1}^{K}$ into a matrix $A \in R^{M*K}$, where M denotes the number of upper triangular elements of matrix $D(k)$. The upper triangular elements of matrix $D(k)$ are pulled into a column vector as the $k$th column of matrix $A$. Each row of matrix $A$ reflects the change of each functional connectivity series $\{\rho_{ij}(k)\}_{k=1}^{K}$ over time. To solve the problem that the correspondence of dynamic FC subnetworks from the same
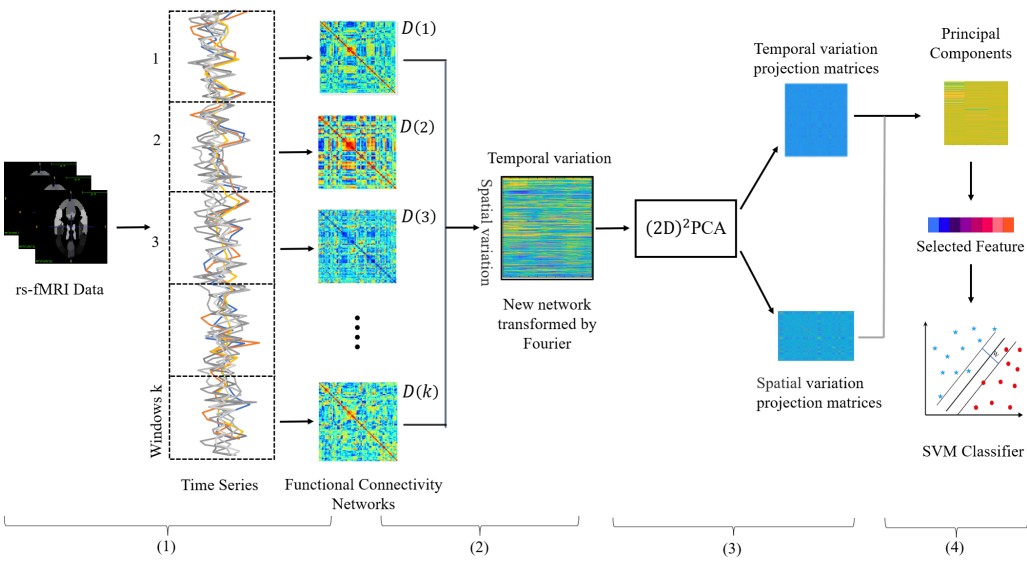

**Figure 1 Illustration of the proposed feature extraction framework integrating both temporal and spatial properties of DFCs.** Including (1) rs-fMRI image preprocessing and networks construction, (2) doing Fourier transform and new network construction, (3) obtaining the best projection matrix in both directions by $(2D)^2$ PCA, (4) Selecting features and training SVM classifier.

window of different subjects cannot be established, we convert the time series $\{\rho_{ij}(k)\}_{k=1}^{K}$ from the time domain to the frequency domain by Fourier transform, which is calculated by Eq. (3). After transformation, the transformation of functional connectivity with time is represented by a linear combination of 'trigonometric functions of different frequencies, so we can analyze the problem from a new perspective, which eliminates the problem of time not corresponding.

Furthermore, we use $(2D)^2$PCA to learn the best projection matrix, which is calculated by Eqs. (5) and (8). The new network is projected by Eq. (9) with the optimal projection matrix to select features that integrate spatial and temporal properties of the new network. $(2D)^2$PCA is widely used for image matrix dimensionality reduction in face recognition and palmprint recognition, which can learn two optimal matrixes from a set of training matrixes reflecting information between rows and between columns of training matrixes, so $(2D)^2$PCA can integrate properties in both directions.

Finally, the features extracted by $(2D)^2$PCA will have high dimensionality, so an effective feature selection method is conducive to accurate diagnosis. For dichotomous classification tasks, $t$-test can be effective in selecting significantly different features between subjects and healthy controls. Therefore, we use a $t$-test to select the features with $p$-values less than a certain threshold value.

## EXPERIMENTS AND RESULTS

This section contains three sub-chapters: methods for comparison, experimental setting and classification performance.

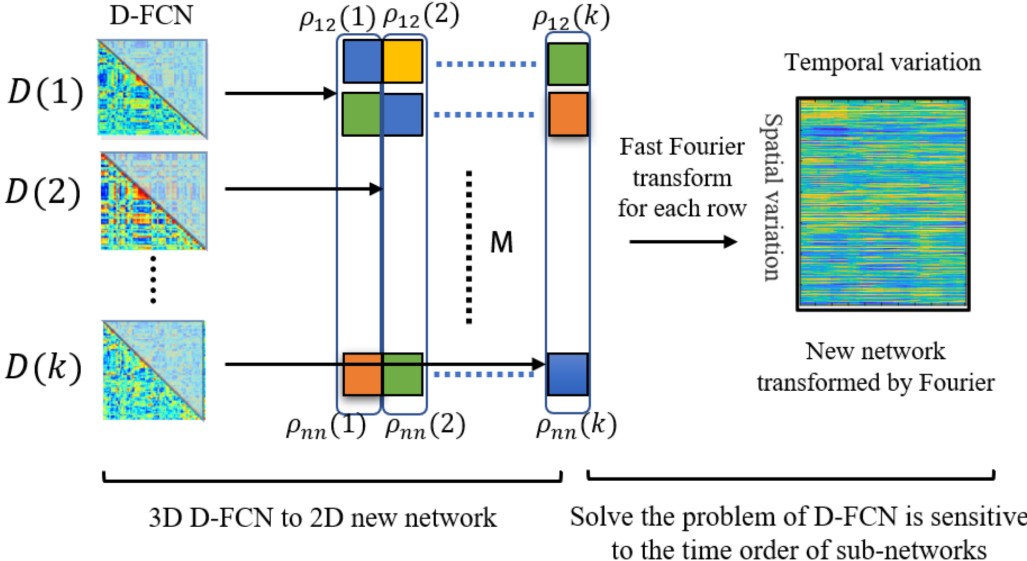

**Figure 2 Illustration the process of new network construction.** $p_{ij}$ (k) denotes the value of the $i$-th row and $j$-th column of functional connectivity network $D(k)$. M denotes the number of upper triangular elements of functional connectivity networks D.

## Methods for comparison

We first compare our proposed method with the central moment method (denoted as Central). In the central moment method, the Pearson correlation between ROI based on the time series of each window is first computed, then the brain networks between ROI are constructed, and the multistep central moment of the brain networks of all the windows solved are used as features for the classification task. Then we compare our proposed method with the method that use topological index of functional connectivity to classify (defined as T-index). In the topological index method, the topology is generated by retaining as nodes the functional connections in the dynamic functions that are greater than a certain threshold, and then the topological index of the topology is derived for binary classification.

Moreover, we compare our proposed method with two methods for extracting unidirectional information using two-dimensional principal component analysis (2DPCA) (*Wang, Wang & Yan, 2018*; *Wee et al., 2016*; *Xu et al., 2019*), which include (1) the method using 2DPCA in the temporal direction (denoted as Fourier-temporal), and (2) the method using 2DPCA in the spatial direction (denoted as Fourier-spatial). In this way, it is demonstrated that $(2D)^2PCA$ can integrate temporal and spatial information improves the performance of classification. It is worth noting that both methods use the Fourier transform to eliminate the effects of time mismatch.

In order to demonstrate that the Fourier transform can play a role in eliminating the time mismatch problem. We also compare our proposed method with that of $(2D)^2PCA$ which does not use Fourier transform (denoted as Temporal-spatial). All the methods use

the same $T$-test as the proposed method for feature selection, and then the features are fed into a SVM classifier for classification.

## Experimental setting

In the experiment, we constructed two classification tasks, including (1) the classification task for autism spectrum disorder (ASD) and normal control group (NC), and (2) the classification task for end-stage renal disease (ESRD) and NC. We used six evaluation indexes, namely, classification accuracy (ACC), true positive rate (TPR), true negative rate (TNR), positive predictive value (PPV), negative predictive value (NPV), and F1 score, to comprehensively evaluate the advantages and disadvantages of the parameters and the classification effect of the methods. Considering that the number of samples used in the experiments is more than 50 and the parameters such as sliding window length, sliding step size, information content thresholds for temporal and spatial directions, and $p$-value thresholds need to be adjusted in the feature extraction framework proposed in this article. After referring to the relevant literature (*Shi et al., 2019*), we selected the ten-fold cross-validation method, which appears in the literature, and did not choose the leave-one-out method, which is more computationally intensive. To avoid possible errors in the experiments, the whole ten-fold cross-validation was repeated 10 times, dividing the samples randomly each time. Finally, the average of the 10 times of recognition accuracy was used as the final accuracy.

In the construction of the network, in order to avoid erroneous experimental results with arbitrarily determined window width and step size, we constructed brain networks with different window widths and step sizes, window width parameter $T \in [20, 30, 40, 50, 60]$ and sliding step size $S \in [2, 4, 6, 8, 10]$. Studies have shown that window width and step size in this range produce robust results in image acquisitions and topological properties of brain networks. We take the DFC derived from all the different parameters and take the average value as the final DFC. In the feature selection phase, we choose features with statistical significance $p < 0.01$. After that, the features are fed into a SVM classifier for training and classification, and the six metrics mentioned in the previous paragraph are calculated (*Yang et al., 2022*; *Yao, Becker & Kendrick, 2021*).

## Classification performance

Table 1 summarizes the classification performance of each of the six methods in the two classification tasks. From Table 1, we can see that our proposed method performs better than comparison methods. For example, the proposed method acc achieves 88.1% and 76.1% in the two classification tasks and the highest accuracy of the comparison method is at 86.1% and 72.8%. Table 1 shows that the method integrating temporal and spatial information of the network performs better than the method using one attribute alone, which implies that the temporal and spatial information are two complementary pieces of information, so integrating the brain through $(2D)^2$PCA network's two kinds of information can further improve the classification performance.

In addition, we could observe that, from Table 1, that using Fourier transform to transform the DFC from the time domain to the frequency domain has a higher

**Table 1  Results of five methods on two classification tasks.**

| Task | Method | ACC (%) | TPR (%) | TNR (%) | PPV (%) | NPV (%) | F1 (%) |
|---|---|---|---|---|---|---|---|
| NC *vs.* ESRD | Fourier-spatial | 85.2 | 88.9 | 81.5 | 84.3 | 89.6 | 85.8 |
| | Fourier-temporal | 85.1 | 88.4 | 81.7 | 83.9 | 87.1 | 86.0 |
| | Temporal-spatial | 86.1 | 84.6 | 87.8 | 88.0 | 84.0 | 86.3 |
| | T-index | 82.2 | 80.8 | 83.7 | 84.0 | 80.4 | 82.4 |
| | Central | 85.1 | 82.4 | 87.7 | 87.6 | 83.5 | 84.7 |
| | Proposed | **88.1** | **92.4** | 84.0 | 86.9 | **91.8** | **89.0** |
| NC *vs.* ASD | Fourier-spatial | 71.7 | 66.7 | 76.7 | 73.2 | 70.1 | 69.8 |
| | Fourier-temporal | 70.1 | 68.9 | 72.3 | 70.0 | 70.1 | 69.7 |
| | Temporal-spatial | 72.8 | 71.1 | 74.5 | 72.7 | 72.9 | 71.9 |
| | T-index | 68.3 | 67.3 | 69.4 | 70.0 | 66.7 | 68.6 |
| | Central | 68.5 | 64.4 | 72.3 | 69.1 | 68.0 | 66.7 |
| | Proposed | **76.1** | **73.3** | **78.7** | **76.7** | **75.5** | **75.0** |

**Notes.**
Values highlighted in bold show best results in the classification task.

classification accuracy than the method that does not use Fourier transform, so we can think that Fourier transform can eliminate the effect of time mismatch to a certain extent and bring better classification performance.

In the process of constructing the dynamic brain network, we used the "sliding window" strategy, so there are two hyperparameters, window width and step size, and we subsequently evaluated the effect of hyperparameters on the classification performance in the classification experiments of nephropathic and normal human beings. Figure 3 shows the effect of different hyperparameters on the classification performance in the four methods, namely Fourier-spatial, Fourier-temporal, Central and our proposed method. It can be concluded that there is an effect of hyperparameters on the classification performance, and our proposed method has more stable and higher classification performance on different combinations of hyperparameters, and has the highest classification accuracy of 88.1% when $T = 50$ and $S = 8$.

## DISCUSSION

Numerous studies have shown significant changes in dynamic functional connectivity in patients with brain diseases. Studying changes in the brain's dynamic connectivity network may contribute to a better understanding of brain diseases and help with early prevention and later treatment. Currently, the study of dynamic functional connectivity in patients with brain diseases is limited to a single feature of a single characteristic. These features do not integrate the temporal and spatial properties of dynamic functional connectivity. In this article, we propose a feature extraction framework that integrates both temporal and spatial bidirectional information of dynamic functional connectivity. Experimentally, it is found that the features extracted by this framework are helpful for the assisted diagnosis of brain diseases, compared to feature extraction methods that contain only temporal direction information.

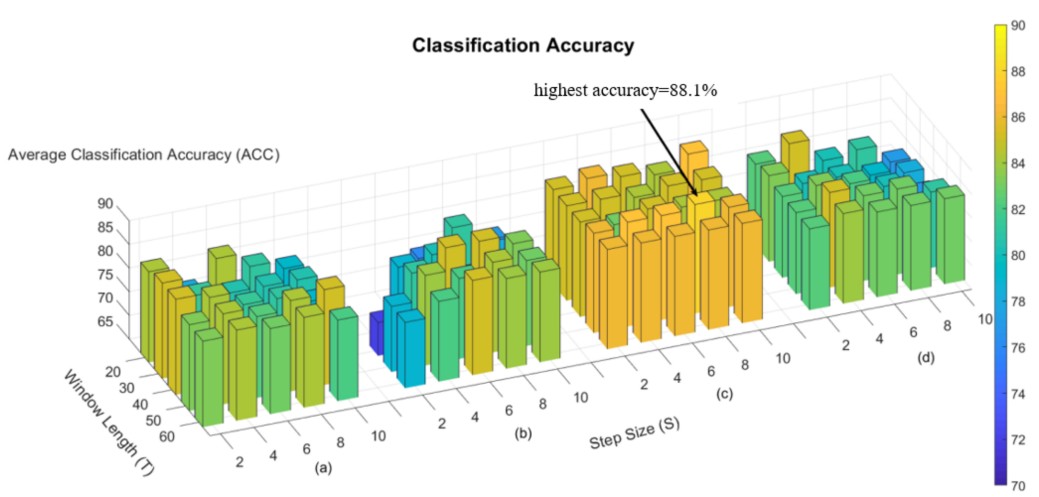

**Figure 3  Accuracy of four methods in the task for ESRD and NC.** (A) ACC of Fourier-spatial. (B) ACC of Fourier-temporal. (C) ACC of our proposed method. (D) ACC of Central.

In order to verify that the features extracted from our proposed framework perform better on the classification task, we used them to train classifiers and make predictions. Considering the two current problems of dynamic functional connectivity mentioned in the article, which may be common to most neurological disorders. In order to validate that our proposed framework has a role to play in a wide range of neurological disorders, we conducted classification experiments on two different neurological disorders datasets. The experiments revealed that the classification performance was improved to some extent on both classification tasks. On the classification task of ASD and NC, the classification performance is improved more, probably because the diagnosis of ASD patients relies more on the combination of temporal and spatial information.

Experiments show that our proposed method has better classification performance than both the central moment method and the local variable method. This suggests that both temporal and spatial information of dynamic functional connectivity can provide support for classification, and combining complementary spatial and temporal information is responsible for the improved classification performance. The classification performance of using 2DPCA for feature extraction framework in one direction is lower than that of our proposed framework, which also confirms that combining complementary spatial and temporal information can improve the classification performance. Also, through experiments we can find that using Fourier transform can improve the classification performance. This is because the Fourier change converts dynamic functional connectivity to frequency domain information, and replaces the way of comparing information at the same time point with the way of comparing information at the same frequency, which avoids the problem of mismatch of information at the same time point and improves the classification performance.

We investigate the important features extracted through our proposed method to discover the brain regions most likely to be abnormal in ESRD patients and autistic

**Table 2  Abbreviations and name of ROIs selected for the experiment.**

|  | Abbreviation | ROI name | Abbreviation | ROI name |
|---|---|---|---|---|
| ESRD | INS | Insula | SMA | Supplementary motor area |
|  | TPOsup | Temporal pole: superior temporal gyrus | ROL | Rolandic operculum |
|  | PoCG | Postcentral gyrus | IPL | Inferior parietal, but supramarginal and angular gyri |
|  | STG | Superior temporal gyrus | PCUN | Precuneus |
|  | CAL | Calcarine fissure and surrounding cortex | ACG | Anterior cingulate and paracingulate gyri |
|  | SFGmed | Superior frontal gyrus, medial |  |  |
| ASD | ITG | Inferior temporal | PCUN | Precuneus |
|  | PHG | Parahippocampal gyrus | TPOmid | Temporal pole: middle temporal gyrus |
|  | REC | Rectus gyrus | TPOsup | Temporal pole: superior temporal gyrus |
|  | PCG | Posterior cingulate gyrus | INS | Insula |
|  | SMA | Supplementary motor area | ORBsupmed | Orbitofrontal cortex (medial) |

patients. Each selected feature, contains both temporal and spatial information, and there exists an element that contributes the most information to that feature in both spatial and temporal directions. In the spatial direction, the element that contributes the most information represents the functional connectivity between two brain regions that we believe are most likely to have abnormalities. For the categorization experiments, the features selected in each cross-validation experiment were different, so we chose features with a $p$-value of less than 0.01 and appearing in each cross-over experiment. The ROIs most likely to be abnormal in patients with end-stage renal disease and autism are given in Table 2, Figs. 4 and 5 depicts these functional connections and the corresponding brain regions, where L and R denote the left and right hemispheres of the brain.

From Fig. 4, we find that the precuneus and temporal lobes dominate among the brain regions most associated with ESRD. Of these, the precuneus is associated with many high-level cognitive functions, such as situational memory, self-related information processing, and various aspects of consciousness, and abnormalities in the precuneus may contribute to the increased risk of cognitive impairment in ESRD patients. The temporal lobe is primarily responsible for language function and auditory perception, may be associated with decreased communication skills in ESRD patients (*Ye et al., 2023*; *Yu et al., 2020*; *Zhang et al., 2011*). From Fig. 5, it can be concluded that most of the brain regions associated with ASD are related to motor coordination and emotional expression, such as the posterior cingulate gyrus (*Zhang & Small, 2006*; *Zhao et al., 2020*; *Zhou et al., 2019*; *Zou et al., 2021*).

This study needs to consider the following limitations. (1) the relatively small sample size in this study will affect the stability of the experimental results. In future studies we will use a larger sample size to verify the correctness of our conclusions. (2) The scan period of rs-fMRI is short, and the dynamic characteristics are not reflected enough. A longer scan period can better reflect the change of functional connectivity over time, which may be crucial for future studies of dynamic functional connectivity. (3) The eigenvalues extracted

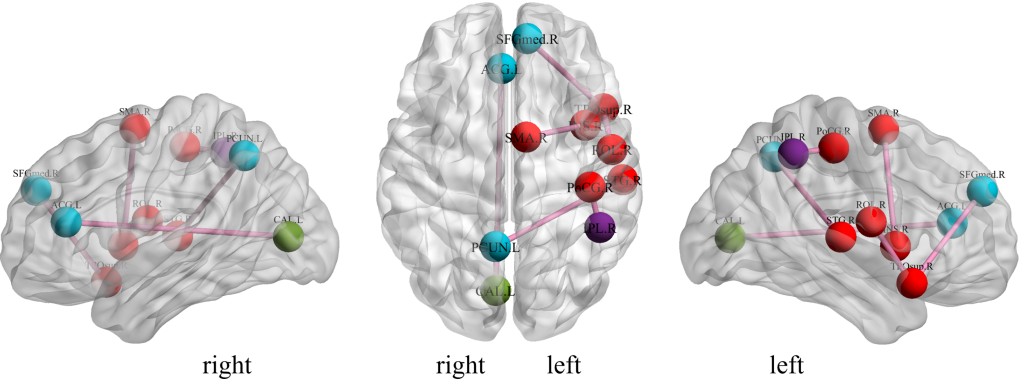

**Figure 4** Diagrammatic representation of the relevant functional connections of important features in ESRD.

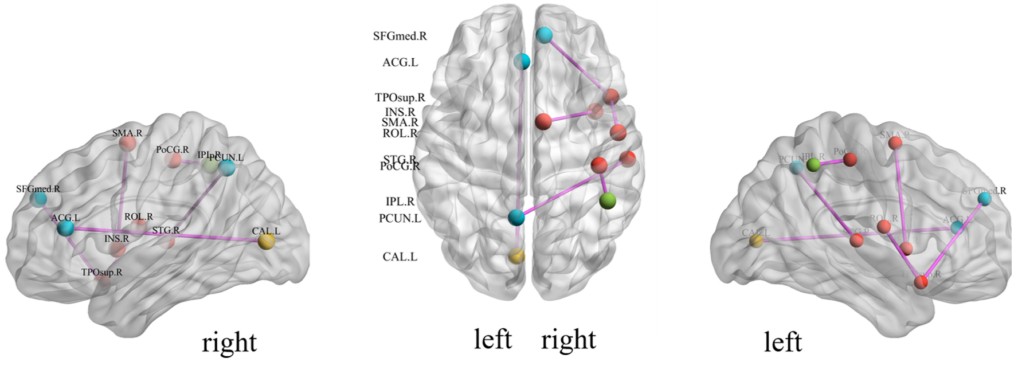

**Figure 5** Diagrammatic representation of the relevant functional connections of important features in ASD.

by the framework are not given a specific meaning and cannot reflect the relevant changes directly through the eigenvalues.

## CONCLUSION

In this article, we propose a feature extraction framework based on $(2D)^2PCA$ that integrates both temporal and spatial properties of dynamic functional connectivity. The framework involves constructing the dynamic connectivity network using a sliding window strategy, addressing the sensitivity of dynamic functional networks to the order of subnetworks through Fourier transform, and leveraging $(2D)^2PCA$ to integrate temporal and spatial information of dynamic functional connectivity. Experimental results demonstrate the complementary effects of considering both temporal and spatial orientations in improving feature extraction performance. These findings provide novel insights into the pathophysiological mechanisms underlying cognitive deficits from a dynamic functional connectivity perspective.

### Funding

This work was supported by the National Natural Science Foundation of China (62176140, 82001775, 61972235, 61976124, 61873117, 61976125) and the Central Guidance on Local Science and Technology Development Fund of Shandong Province (YDZX2022093). There was no additional external funding received for this study. The funders had no role in study design, data collection and analysis, decision to publish, or preparation of the manuscript.

### Grant Disclosures

The following grant information was disclosed by the authors:
National Natural Science Foundation of China: 62176140, 82001775, 61972235, 61976124, 61873117, 61976125.
Central Guidance on Local Science and Technology Development Fund of Shandong Province: YDZX2022093.

### Competing Interests

The authors declare there are no competing interests.

### Author Contributions

- Feng Zhao conceived and designed the experiments, performed the experiments, analyzed the data, prepared figures and/or tables, authored or reviewed drafts of the article, and approved the final draft.
- Ke Lv performed the experiments, analyzed the data, prepared figures and/or tables, authored or reviewed drafts of the article, and approved the final draft.
- Shixin Ye analyzed the data, authored or reviewed drafts of the article, and approved the final draft.
- Xiaobo Chen analyzed the data, authored or reviewed drafts of the article, and approved the final draft.
- Hongyu Chen analyzed the data, authored or reviewed drafts of the article, and approved the final draft.
- Sizhe Fan analyzed the data, authored or reviewed drafts of the article, and approved the final draft.
- Ning Mao analyzed the data, authored or reviewed drafts of the article, and approved the final draft.
- Yande Ren analyzed the data, authored or reviewed drafts of the article, and approved the final draft.

### Data Availability

The code is available at GitHub and Zenodo:
- https://github.com/LvshanD/Code/tree/papercode
- Lv. (2023). code. Zenodo. https://doi.org/10.5281/zenodo.10443810

The Autism Brain Imaging Data Exchange (ABIDE) database is available at:
http://fcon_1000.projects.nitrc.org/indi/abide/abide_I.html.

The functional connectivity raw data for the experimental patients with end-stage renal disease is available in the Supplemental File.

## Supplemental Information

Supplemental information for this article can be found online at http://dx.doi.org/10.7717/peerj.17078#supplemental-information.

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
