# Peer review of "Integration of temporal & spatial properties of dynamic functional connectivity based on two-directional two-dimensional principal component analysis for disease analysis"

_PeerJ, doi:10.7717/peerj.17078_

## Round 0.1 · original submission · Major Revisions

All three reviewers raised major concerns, and the authors should carefully address them.

Reviewer 1 ·

Basic reporting

The feature extraction method and the method for mitigating time mismatch presented in this paper are innovative to some extent. I agree to publish this paper, but need the authors to revise or explain the following points:
1.The expressions of many sentences need to be improved, such as: Line45-47; Line 206-207, ect.
2.The font sizes in form Table 1 is not uniform.
3.In line 178, “The matrix C in Eq. (10) is the matrix after projection.” Eq. (10) is not defined.

Experimental design

1.The authors used the t-test to select features. Please give a detailed explanation on the reason why the method is used.
2.Ask the authors to explain why 92 subjects were selected from the ABIDE database?

Validity of the findings

no comment

Additional comments

1.Add more information about the current state of research in the introduction section to improve the quality of manuscript.
2.The abstract of this article needs to be reorganized.

Reviewer 2 ·

Basic reporting

no comment

Experimental design

no comment

Validity of the findings

no comment

Additional comments

Understanding and predicting mental illness is an important research issue. This paper proposes a new feature extraction framework based on two-directional two-dimensional principal component analysis. The experimental results show that the features extracted by this framework have better results in classifcation experiments than those features containing individual properties. However, the manuscript is poorly written. I have several suggestions and comments to improve this manuscript quality further:
1. I suggest that the author adjust the structure of the introduction and focus the paper on classification.
2. Too few references and irregular citations.
3. The authors used two different datasets but do not explain this in-depth in the Discussion section. Authors may consider using only data collected in their own laboratory.
4. The data preprocessing steps in the manuscript are incomplete, such as time correction, smoothing, quality control, etc.
5. What are the inclusion and exclusion criteria for subjects?
6. The sample size for classification is not large. The author used ten-fold cross-validation. Why is the leave-one-out cross-validation method not applicable?
7. Materials and Methods section, the authors should provide more details. For example, the central method and selection of window length, etc.
8. The discussion is not sufficient, and the advantages of the classification results are not well explained.

Reviewer 3 ·

Basic reporting

This paper introduces a new feature extraction framework for dynamic functional connectivity that integrates temporal and spatial attributes. The Fourier transform is used to alleviate the time mismatch problem in functional connectivity studies. This is an interesting work, and is written well.
Several comments:
1. In Section 3.3, “(2D)^2 PCA is a matrix feature extraction technique based on 2DPCA. 2DPCA can extract whole-matrix features by finding optimal projection directions from the covariance matrix.” Pls provide references for the involved methods. Acronyms should be explained when they are first mentioned.
2. Describe in detail in the discussion section how to derive possible anomalous functional connection from the selected features.
3. Many typos exist in the paper. For exampel, a space is missing between Fig. and 5.

Experimental design

1. In the experimental section, there is a lack of comparative experiments between the features extracted in this article and the features with spatial attributes. A related experiment is strongly recommended.
2. In section 2.1. “The rs-fMRI data of 45 autism spectrum disorders (ASD) patients and 47 healthy controls (HC) groups were obtained from Autism Brain Imaging Data Exchange (ABIDE) database29 that scanned at New York University Langone Medical Center.” As I know, the number of subjects in the ABIDE dataset is much larger than that used in this paper (only 45 ASD and 47 NC). On what foundation did the authors choose them?

Validity of the findings

No comment.

Additional comments

The research motivation needs to be further clarified.

---

## Round 0.2 · accepted · Accept

All the comments have been addressed.

Reviewer 1 ·

Basic reporting

no comment

Experimental design

no comment

Validity of the findings

no comment

Additional comments

no comment

Reviewer 2 ·

Basic reporting

no comment

Experimental design

no comment

Validity of the findings

no comment

Additional comments

The author has answered all my questions very well. I have no more questions.

Reviewer 3 ·

Basic reporting

The authors have revised the manuscript carefully according to my comments.

Experimental design

No futher comments

Validity of the findings

No comment